# A Climate Service for Ecologists: Sharing pre-processed EURO-CORDEX Regional Climate Scenario Data using the eLTER Information System

Susannah Rennie[1], Klaus Goergen[2,3], Christoph Wohner[4,5], Sander Apweiler[6], Johannes Peterseil[4], John Watkins[1]

[1]UK Centre for Ecology & Hydrology, Lancaster Environment Centre, Library Avenue, Bailrigg, Lancaster, LA1 4AP, UK

[2]Institute of Bio- and Geosciences (Agrosphere, IBG-3), Research Centre Jülich, Wilhelm-Johnen-Straße, 52425 Jülich, Germany

[3]Centre for High-Performance Scientific Computing in Terrestrial Systems, Geoverbund ABC/J, 52425 Jülich, Germany

[4]Environment Agency Austria, Spittelauer Lände 5, 1090 Vienna, Austria

[5] Paris-Lodron University of Salzburg, Schillerstraße 30, 5020 Salzburg, Austria

[6]Jülich Supercomputing Centre, Research Centre Jülich, Wilhelm-Johnen-Straße, 52425 Jülich, Germany

Correspondence to: Susannah Rennie (srennie@ceh.ac.uk)

**Abstract**. eLTER was a 'Horizon 2020' project with the aim of advancing the development of long-term ecosystem research infrastructure in Europe. This paper describes how eLTER Information System infrastructure has been expanded by a climate service data product providing access to specifically pre-processed regional climate change scenario data from a state-of-the-art regional climate model ensemble of the COordinated Regional Downscaling EXperiment project (CORDEX) for 702 registered ecological research sites across Europe. This tailored, expandable, easily accessible dataset follows FAIR principles and allows researchers to describe the climate at these sites, explore future projections for different climate change scenarios and make regional climate change assessments and impact studies. The data for each site are available for download from the EUDAT Collaborative Data Infrastructure B2SHARE service and can be easily accessed and visualised through the Dynamic Ecological Information Management System - Site and Dataset Registry (DEIMS-SDR), a web-based information management system which shares detailed information and metadata on ecological research sites around the globe. This paper describes these data and how they can be accessed by users through the extended eLTER Information System architecture.
The data and supporting information are available from B2SHARE. Each individual site (702 sites are available) dataset has its own DOI. To aid data discovery, a persistent B2SHARE lookup table has been created which matches the DOIs of the individual B2SHARE record with each DEIMS site ID. This look up table is available at http://doi.org/10.23728/b2share.bf41278d91b445bda4505d5b1eaac26c (eLTER EURO-CORDEX Climate Service, 2020).

# 1 Introduction

Long-term ecosystem research (LTER) is a key component of global efforts to better understand the structure and functions of ecosystems and their response to environmental, societal and economic drivers (Nolan et al., 2018). This should help to maintain these ecosystems and allow for a sustainable use of ecosystem services. LTER-Europe (LTER-Europe, 2020) is the formal European regional group (Mirtl, 2010) of the global ILTER network (Mirtl, 2018). eLTER was a flagship project for LTER-Europe to advance the development of LTER infrastructure in Europe. It comprised 26 countries and around 450 field sites, where LTER research and monitoring was undertaken. Funded through the Horizon 2020 programme, this project ran from 2015 to 2019 (CORDIS, 2020a) and is continued through another Horizon 2020 project "eLTER Plus" (runtime from 2020 to 2025; CORDIS, 2020b) which aims to develop eLTER as a formally recognised ESFRI Research Infrastructure (RI). A major goal of LTER is to provide long-term, reliable and quality controlled data for scientific analysis as well as the assessment of environmental policy impacts. The eLTER Information System (eLTER Information System, 2020) provides a common information management infrastructure for making environmental data from distributed resources (it's contributing national networks) available and accessible for users in accordance with FAIR principles (Wilkinson et al., 2016). eLTER has designed, implemented and operates a federated data infrastructure (Fig. 1), with data stored in existing partner data systems, harmonised by a central discovery portal and federated data access components. Its components (eLTER Information System, 2020) include:

- A metadata editor and catalogue (DEIMS-SDR, Dynamic Ecological Information Management System – Site and Dataset Registry) which allows users to create, publish and share information on (research) sites, (data collection) activities, datasets and sensors (Wohner et al., 2019; DEIMS-SDR, 2020).

- A common controlled vocabulary (EnvThes) which harmonises the terminology used to describe data shared within eLTER (EnvThes, 2020).

- The Data Integration Portal (DIP) which enables users to discover, visualise and access data from multiple sites and sensors (DIP, 2020).

- A Central Data Node (CDN) which allows users to register time series data services and upload data. Once this upload is completed, these data can be visualised and accessed through Sensor Observation Service (SOS) clients such as the eLTER DIP (CDN, 2020).

The eLTER Information System complies with the EU INSPIRE regulations (INSPIRE, 2020) by providing metadata in dedicated metadata formats and providing data using Open Geospatial Consortium (OGC) standard Web Feature (WFS), Web Mapping (WMS), Web Coverage Services (WCS) and Sensor Observation Services (SOS) as well as dedicated services for certain subsystems (SparQL for the EnvThes; Rest-API for DEIMS-SDR).

The eLTER Information System infrastructure has been expanded to offer a complementary climate service data product that specifically addresses the needs of the ecological research user community working in the LTER context. This climate service provides open access to time series of regional climate projections from the latest state of the art EURO-CORDEX (EURO-

CORDEX, 2020) regional climate model (RCM) ensemble (Jacob et al., 2014; Jacob et al., 2020) for time spans up 1950 to 2100 at the site level for 702 DEIMS-registered ecological research sites located in Europe - this includes all the eLTER sites. These are discoverable through DEIMS-SDR and are freely available for download by users from the EUDAT B2SHARE service (Ardestani et al., 2015). This paper describes these data and how they can be accessed by users through the eLTER Information System architecture.

## 2 Data

The eLTER regional climate scenario data (eLTER EURO-CORDEX Climate Service, 2020) are based entirely on an ensemble of regional climate model (RCM) simulations from the World Climate Research Programme's (WCRP, 2020) Coordinated Regional Downscaling Experiment (CORDEX) (Giorgi et al., 2015; Gutowski et al., 2016; CORDEX, 2020) project and its European initiative EURO-CORDEX (Jacob et al., 2020). CORDEX RCM results are publicly and freely available through a global system of federated data centres which form the largest archive of climate data world-wide, the Earth System Grid Federation data nodes (Cinquini et al., 2014; for example ESGF node, 2020 is one of its access points).

The EURO-CORDEX RCM ensemble data made available through the eLTER information system consist of dynamically downscaled CMIP5 (Taylor et al., 2012) global climate model (GCMs) simulation results for the historical time span up to 2005 based on observed greenhouse gas (GHG) concentrations and simulation results for the future time spans based on different GHG concentration trajectories, the representative concentration pathways (RCPs) (Moss et al., 2010). EURO-CORDEX is one of the most extensive and state-of-the-art RCM ensemble datasets of this kind, dynamically downscaling a wide range of the CMIP5 GCMs and the most widespread RCPs (Jacob et al., 2014; Jacob et al., 2020). Three different RCPs (RCP2.6, RCP4.5 and RCP8.5) are considered, each of them having a different number of ensemble members (i.e., individual RCM runs) available. The current version 1 of the eLTER regional climate scenario data consists of 6 ensemble members for RCP2.6; 14 ensemble members for RCP4.5; and 14 ensemble members for RCP8.5. As the number of EURO-CORDEX ensemble members on the ESGF data nodes grows continuously, the base datasets and the eLTER data product have to be expanded continuously in further dataset versions (see Sect. 10).

EURO-CORDEX RCM simulation results have been used and tested extensively in recent years in various evaluation studies (e.g., Kotlarski et al., 2014; Vautard et al., 2013; Knist et al., 2017; Prein et al., 2016) and regional climate change assessments (e.g., Jacob et al., 2014). The CORDEX ensemble is the most comprehensive coordinated regional climate change dataset generated so far (Gutowski, 2016). Validation studies such as Kotlarski et al. (2014) for the EURO-CORDEX ERA-Interim reanalysis driven evaluation simulations show good agreement with observations. The added value of the 12km model runs, for example in the reproduction of precipitation properties with reference to coarser resolution runs, is assessed in Prein et al. (2016); and regional climate scenario ensemble validation studies, for example to investigate the regional climate change and also changes in extremes over Europe is explored in Jacob et al. (2014). The EURO-CORDEX data are a basis for many national regional climate change assessments (e.g., Kjellström et al., 2016; Rössler et al., 2019).

It should be noted that regional climate modelling downscaling and impact modelling chains are characterised by large inherent total uncertainties. These are due to the RCPs used, the GCM, the RCM (or statistical downscaling), post-processing methods (incl. bias adjustments), and the impact model; the models have structure and parameter uncertainty, bias-adjustment methods may be prone to instationarities of their parameter settings, and natural variability also contributes to total uncertainty (e.g., Bosshard et al., 2013; Christensen and Kjellström, 2020; Her et al., 2019; Kundzewicz et al., 2018). Decomposing the total uncertainty of a GCM-RCM dynamical downscaling modelling chain into its different sources and contributions, for example Christensen and Kjellström (2020) show that the GCM has a large influence on the climate change signal, while the choice of RCM impacts results e.g. in mountainous areas considerably; Sørland et al. (2018) show that modelling chain errors are not additive, but that RCM can reduce biases inherited by the GCM. Littel et al. (2011) provide an overview on the interplay of the different sources of uncertainty in using impact modelling chains in ecology.

Data from CMIP6 (Eyring et al., 2016) GCM-driven RCM simulations will most likely form a further base dataset in a future version of the eLTER regional climate scenario data. Also, data from high-resolution convection-permitting regional climate model experiments at spatial resolutions below 4km show added value through a more realistic reproduction of many meteorological processes (e.g., Prein et al., 2015). But due to their computational cost, such ensemble datasets are still mainly focussing on smaller spatial domains and fewer GCMs, which are dynamically downscaled, such as in the UK Climate Projections (Kendon et al., 2019), or the ensemble experiments are run not only for sub-continental domain, but also for time slices, as in CORDEX Flagship Pilot Study on Convective phenomena at high resolution over Europe and the Mediterranean (Coppola et al., 2018).

The eLTER regional climate scenario data were extracted from an ensemble of EURO-CORDEX RCM simulations for 702 DEIMS-SDR registered European ecological monitoring sites that are covered by the EURO-CORDEX EUR-11 (12 km spatial resolution) RCM focus domain. The coordinates for these sites were obtained from DEIMS-SDR and are shown in Fig. 2. Time series for all these sites were extracted, irrespective of how close they were to each other, from the generic model grid, providing an ensemble of past and future regional climate data for that site (with various limitations and constraints – see Sect. 7).

For each RCP, 8 near surface variables (Table 1) were retrieved from the ESGF data node, and made available per site. These variables are a subset of those available from EURO-CORDEX (more than 60 variables are available; CORDEX variables, 2020) but the eight that were chosen are commonly used to assess regional climate change, extremes and as input data to impact assessment and modelling. The timespan of the RCM simulations used is up to 151 years, from 1950 to 2100. For 1950 to 2005 observed GHG concentrations are used in the control simulations and for 2006 to 2100 the driving CMIP5 GCMs use the GHGs from the RCPs. For ease of use, the daily time series of the historical and the projection time spans are concatenated to form a single, up to 151-year long time series, albeit not all RCMs cover the complete timespan from 1950 to 2100. The spatial resolution of the RCM base data from which the time series are derived is about 12 km. In addition to the original daily temporal resolution as retrieved from the ESGF data nodes, all time series are also provided as monthly and yearly time series. Detailed notes about how to use these data, and their limitations and constraints, are found in Sect. 7.

## 3 Data visualisation and basic diagnostics

Based on the time series data, for each site, RCM ensemble member, RCP, and variable, regional climate change is described through pre-processed descriptive statistics and typical visualisations. These visualisations provide an assessment on long-term mean regional climate change and on the ensemble spread per individual site. The box plots (Fig. 3) show and example of seasonal and annual near (2021-2050) and far (2069-2098) future changes with reference to the historical time span (1971-2000). These visualisations are made available through DEIMS-SDR (see Sect. 5).

**4 Data storage**

Pre-processed RCM data are provided via the EUDAT B2SHARE service (B2SHARE, 2020). B2SHARE provides researchers a solution for storing and sharing small-scale research data from diverse contexts. It can handle research data in a single file and distributed data across multiple files. The research data and additional metadata are stored in one record. While creating a record the user is able to fill in the (community specific) metadata.

B2SHARE is a long-term, persistent open-access research data storage infrastructure hosted at the Jülich Supercomputing Centre (JSC, 2020), one of the three national Supercomputing centres in Germany. Each submitted B2SHARE record and each file within the record is provided with a PID (persistent identifier). The record is additionally provided with a DOI. Changes in the uploaded data are versioned and every version has its own PID and DOI.

The time series data are made available on an individual file basis, i.e., per site, variable, RCP, temporal aggregation, file type, 145 and RCM ensemble member. This detailed level of granularity was chosen to allow maximum flexibility for all usage scenarios and also allow for an expansion of the dataset in terms of number of variables, number of ensemble members and number of sites in the future. A guiding principle was the requirement of the eLTER data users for site-specific, detailed data that are easy to use and integrate into existing analysis or modelling workflows.

The basic file format used is netCDF4; the nc files contain the unaltered original metadata following the CMOR standard for 150 data provenance tracking plus a global history attribute which provides a record of the processing. In addition, basic ASCII text files with tabulated data (format: year, month, day, time, value) are also provided, which allow for an immediate import into spreadsheet software tools. For studies which require data from many sites, combined site and ensemble data in a single netCDF may provide better usability. To avoid having too many files per site; to improve the efficiency of data handling, retrieval and storage; and to prevent users from selecting individual ensemble members instead of the full ensemble provided, 155 the ensemble members per variable (x8), RCP (x3), temporal aggregation (x3) and file type (x2) are stored together in a compressed zip archive. Each zip file contains the same pdf file with supporting information on the data, advice on how to use them, a discussion on their limitations, contact details, versioning and licensing information, a change log, and a disclaimer. Hence, altogether 144 zip archive files are provided per site and for each site one B2SHARE record, a so-called deposit, is created (Fig. 4). As files are automatically staged, each record is assigned a unique B2SHARE identifier (PID) and a DOI. 160 These B2SHARE records are also automatically versioned.

DEIMS-SDR creates persistent and unique identifiers for each site using Universally Unique Identifiers (UUID; Version 4). A persistent, resolvable and unique identifier (DEIMS.ID) for each site is created using the base url of DEIMS-SDR and the issued UUID. These DEIMS.IDs have been integrated into the data processing workflow for the regional climate scenario datasets. They are included in the file names of the zip files containing the ensemble member files, the individual data files, the graphics file names and the visualisations to ensure that these files are always clearly attributed to the correct monitoring site. All data files within the zip archive file follow the CMOR-based data reference syntax from EURO-CORDEX plus the site UUID and encoded processing information. This naming scheme ensures that file names are always unique (example available in Fig. 4). Even with individual ASCII files, which do not contain any metadata, data provenance can be derived. This strict naming convention allows for an efficient usage of the large number of files through automatically generated file lists, however it does create long file names (see Sect. 7 for information on how to handle these).

## 5 Data dissemination

The eLTER RCM datasets can be found through either a search on B2SHARE or via DEIMS-SDR - using DEIMS-SDR is the preferred mode of access. DEIMS-SDR is a web-based portal describing a wide range of sites including their location, facilities, research themes and data availability around the globe (Wohner et al., 2019). A DEIMS-SDR site search (DEIMS-SDR, 2020) allows users to discover information about the site and view the climate scenario data visualisations for the site (Fig. 5) and links to download these data from B2SHARE. The DEIMS-SDR site map (DEIMS-SDR site map, 2020) (Fig. 6) can also be used to explore the data. Users can select a site of interest from this map viewer and click 'Show more details' to view a pop-up window that provides links to the visualisations and data from B2SHARE, as well as information about the site and other data that might be available. The current dataset is released as version 1. Further releases are planned, which could expand the number of ensembles available and further technical properties (see Sect. 7 and Sect. 10).

Given the number of datasets available (data are available for 702 individual sites), a persistent B2SHARE lookup table has also been created to aid data discovery which matches the DOIs of the individual B2SHARE record with each DEIMS site ID (eLTER EURO-CORDEX Climate Service, 2020).

For bulk retrievals, it is possible to harvest data from B2SHARE via wget or by using the B2SHARE REST-API. The overall metadata of a B2SHARE record, fetched via the REST API, contains a link to the file bucket of the record. The metadata of the file bucket contains information such as the mimetype, URLs of the files and checksums. The files can be downloaded by the file URLs with common tools, e.g. wget. After downloading the files the data can be validated, to avoid data corruption, using the md5-checksums.

## 6 Benefits for LTER community

These regional climate scenario data are intended for the ecological research community, providing easy access to climate scenario data and visualisations for general regional climate change assessments or for impact assessments. Selecting the parts of the EURO-CORDEX data that a researcher may need and formatting the data correctly is time-consuming. It also requires knowledge and skills that many ecologists do not have. Access to pre-processed and cookie-cut model data, through the eLTER Information System, for their sites of interest will save LTER researchers time, and for some researchers it will provide

access to data resources that they will not previously have had; increasing their capabilities for long term ecological research. The eLTER regional climate change scenario data have already been used in ecological impact studies (e.g. Dirnböck et al. (2018); Holmberg et al. (2018)). In Dirnbock et al. (2018) the site-based regional climate change scenario and deposition data were used in connection with a dynamic soil model coupled to a statistical plant species niche model to explore expected plant response to legislated reductions in nitrogen emissions. Holmberg et al. (2018) looked at the impacts of deposition and climate

change on soil conditions, using the regional climate change scenario data to explore long-term ecosystem impacts.

By making a large ensemble of regional climate change information discoverable and accessible to the eLTER, and the wider ecological research community, through the eLTER Information System (with its links to the site information and the datasets) the utility of these datasets for ecological research is much improved. DEIMS-SDR is widely used within the LTER community for the provision and search of site-based metadata. B2SHARE was the recommended repository for data to be

submitted to the eLTER Information System during the H2020 project and it will continue to be important during the development of the eLTER RI. Embedding the regional climate change scenario data within the systems (DEIMS-SDR and B2SHARE) that are already familiar to LTER researchers for searching metadata on sites, and storing datasets collected from those sites, should be beneficial and, assuming the eLTER RI is successfully established, available to researchers for an extended period of time. The combination of providing site metadata, the observed datasets and the pre-processed climate

modelling data for each site through the eLTER Information System is a unique resource for LTER researchers.

Additionally, closer links to observed data from eLTER sites should help improve the regional climate change scenario data - as noted in Dirnbock et al. (2018) LTER sites are useful "reference systems for developing and validating ecological models." It is anticipated that environmental policy development will become increasingly reliant on research infrastructures like the eLTER RI and the integrated ecosystem models they are enabling (Mirtl et al. 2018), streamlining the data and models into

the eLTER Information System should make this easier.

## 7 Technical information and limitations

To allow data users to evaluate whether the dataset and the associated service is applicable for their intended usage scenario, processing details, technical properties as well as limitations, uncertainties and constraints are described in this section. This information is also provided in each data granule in a disclaimer when the zip files are downloaded from B2SHARE. A guiding

principle in the production of the eLTER regional climate scenario data was to add as little additional uncertainty to the data product due to the processing as possible.

All data processing has been done using the Climate Data Operators (CDO, 2020) v1.9.1, developed and provided by the Max Planck Institute for Meteorology (Max Planck Institute for Meteorology, 2020). The CDO processing information is provided in the history global attribute in each of the netCDF files.

No bias adjustment (Maraun, 2016) has been applied to the time series. This will be considered and included in the service for selected sites in the future based on meteorological observations of those sites, where this data can be made available (not all the sites make meteorological observations). At the time of the generation of version 1 of the eLTER climate service datasets, these site data were not yet included in the DEIMS-SDR. Especially with respect to threshold-related processes or climate change indices, not having a bias adjusted time series is a limitation (Hoffmann et al., 2018). However, Casanueva et al. (2020)

point out the importance of suitable reference observations; given the large uncertainties of observational reference data (Kotlarski et al., 2018; Herrera et al., 2019; Prein and Gobiet, 2017) and considering the resolution mismatch between a gridded product and the LTER sites, we refrained from using a non-site based meteorological time series as the basis for a bias adjustment.

Additionally, no height correction has been applied to adjust the air temperature, e.g. via a constant lapse rate of 0.65K/100m,

from the altitude of the grid cell which corresponds to the model output to the altitude of the site. In areas with steep and highly variable topography, this can introduce substantial deviations between the model's topography and the real altitude at the site. If a user wants to use eLTER regional climate scenario data for such a site, this might be a strong limitation for the applicability of the data. Snell et al. (2018) address the sensitivity of a forest landscape model to different climate datasets, height corrections and downscaling methods over a mountainous area.

The time series are extracted from the original EURO-CORDEX RCM model grid using a bilinear resampling. This means that all four surrounding grid points of a site are considered for the time series generation (i.e., depending on the respective distance of each of the model grid point centres to the site, the RCM grid points are weighted when calculating the spatial mean from raw model output daily data). Using a four-point neighbourhood of the LTER site to derive the time series introduces a smoothing but can also help to compensate for mismatches in land cover in the RCM and the local LTER site,

which has more influence on near surface meteorological fields in case of a nearest neighbour time series extraction.

The land-ocean mask, i.e., whether a model grid point on the RCM grid is located over land or ocean, is deliberately not considered; all four neighbouring RCM grid points are always used to calculate the site time series. The underlying assumption is that the site is so close to an ocean or inland water body grid point in the RCM, this proximity would, in the real climate system, also have an impact on the measurements at the site therefore such grid points are also considered. This might happen

at sites close to larger lakes or close to the coastline but may not be indicated in the time series datasets.

The time series for the historical (1950 to 2005, based on observed GHGs) and the projection time spans (2006 to 2100, based on the climate scenario RCP2.6, RCP4.5, RCP8.5 GHGs) have been merged into a single time span. This makes using the individual time series easier, e.g., when calculating running climate indices and so forth. A downside of this is that if there are,

e.g., two RCPs, 4.5 and 8.5, that a specific GCM has been run with, the historical timespan until 2005 is run only once. The RCM, which is downscaling the GCM also follows this scheme. Hence, the extracted time series until 2005 would be identical between the two C20-RCP4.5 and C20-RCP8.5 time series. The overall temporal coverage of the time series may vary, i.e. some RCM simulations may start later than 1950 and some may end earlier than 2100-12-31. In addition, depending on the model system, a standard calendar is not always used (i.e., 365 days and including leap years). Users are advised to use processing tools which take account of the date and time information of the netCDF files, such as the Climate Data Operator tools.

The RCM ensemble, as retrieved and checksum-verified from ESGF, has not been altered when pre-processing data. Also no additional checks, compliancy, plausibility or quality checks are done with the raw RCM data. Users should take this into account when analysing data for a specific site. The version 1 dataset contains only EURO-CORDEX RCMs from dynamically downscaling CMIP5 r1i1p1 GCM ensemble members (see Sect. 8 for updates). It must be noted that the regional climate scenario data are solely based on dynamical downscaling of CMIP5 GCMs, as used in Jacob et al. (2014). If detailed evaluation of the RCM results in relation to meteorological observations is to be performed, the ERA-Interim reanalysis driven ensemble should be used instead e.g. as in Kotlarski et al. (2014).

A very detailed guideline on the limitations, specific properties and the use and interpretation of regional climate scenario ensemble dataset is provided by the EURO-CORDEX initiative (EURO-CORDEX guidelines, 2020). Kreienkamp et al. (2012) also provides a guideline for best practice. The limitations and constraints of the data product needs to be considered before drawing any conclusions and/or using the data in further work.

Because the archive files and data file filenames can be quite long, problems might occur when unpacking data due to path length limitations in Windows. To circumvent this issue: unzip the data at a higher directory level to avoid unnecessarily long path names. The long filenames are used to ensure clear identification of the data and to associate the data unambiguously with the respective sites.

## 8 Datasets in context

The intention of offering the eLTER climate service data product is to specifically address the needs of the ecological research user community. Providing access to pre-processed, site-based data through the eLTER information system should save ecologists time and will open up these data to researchers who may not have the skills to work with RCM data directly. DEIMS-SDR and B2SHARE offer open and simple access to data for ecological researchers. The intention for this section is to provide context for those ecological researchers who may not have used or had access to regional climate scenario data before. It should be noted that some of the data resources included in this section may not have an open data policy but users can freely access the data through the eLTER Information System as described in this paper.

Regional climate change has multiple effects on natural and human systems through changes in the physical system and the abiotic drivers, impacting ecosystems, their functioning and services (Diffenbaugh and Field, 2013; Nolan et al., 2018; Runting

et al., 2017). Europe's future climate evolution is projected to be characterised by regionally varying air temperature and precipitation changes (Kovats et al., 2014; Jacob et al., 2014). Climate projections show for example increases in long-term mean European air temperature, as well as contrasting regional climate change signals of decreasing precipitation in Southern Europe and increasing precipitation in the North. These changes could trigger ecological responses in plants and animal communities.

Regional climate models have a number of advantages over global climate models in reproducing regional features of the climate system and producing data for applications (Giorgi 2019; Rummukainen, 2016). Regional climate change impacts on ecosystem functioning or services are highly relevant for the eLTER network (Diffenbaugh and Field, 2013; Nolan et al., 2018; Hoegh-Guldberg et al., 2019; Runting et al., 2017; Holmberg et al., 2018; Dirnboeck et al., 2018). To make specific use of climate change information across sectors and help distil the ever increasing data volumes from climate models (e.g., Overpeck 2011) climate services (Hewitt et al., 2012) have evolved, which can be classified into different types (Visscher et al., 2020). Overviews of existing climate information or services are available (Hewitt et al., 2017) and more recently specifically for Europe by Cortekar et al. (2020).

Beyond the eLTER information system, and the regional climate scenario data that were specifically prepared and maintained for the eLTER community and related research initiatives, a large number of climate services with a focus on Europe exist (Cortekar et al., 2020), with a large differentiation among different sectors (Bruno Soares et al., 2018). Data portals that also make EURO-CORDEX data more easily accessible than a direct ESGF retrieval are, e.g. Climate4Impact, 2020 and the Copernicus Climate Data Store (CDS, 2020; Buontempo et al., 2020), albeit not tailored to the eLTER or ecological community.

## 9 Data availability

CORDEX regional climate data are available as open access research data (CORDEX, 2020) and the EURO-CORDEX data that have been used to generate the regional climate scenario data for the eLTER project are free for non-commercial use. If any commercial use is intended, the user has to make sure that the respective RCM ensemble members are under an unrestricted terms of use, as indicated on the CORDEX website. An identification of each dataset is possible due to an unambiguous data identification through the file metadata (netCDF) and the filename (netCDF and ASCII). The B2SHARE and DEIMS-SDR services are available free of charge for searching and download.

Each individual site (702 sites are available) dataset has its own DOI, available on the relevant B2SHARE record, and the regional climate scenario datasets can be cited using these. To aid data discovery, a persistent B2SHARE lookup table has been created which matches the DOIs of the individual B2SHARE record with each DEIMS site ID. This look up table is available at http://doi.org/10.23728/b2share.bf41278d91b445bda4505d5b1eaac26c (eLTER EURO-CORDEX Climate Service, 2020). If new datasets are released, these will be made available on the relevant B2SHARE record. B2SHARE datasets are versioned and every version has its own PID and DOI.

## 10 Conclusions and outlook

The eLTER regional climate projections form a multi-functional regional climate scenario data repository for the ecological research community using the EURO-CORDEX state of the art RCM ensemble as base data. It has expanded the eLTER Information System following FAIR principles by making a range of RCM climate change scenario datasets and analysis easily accessible and available on an individual site basis. This new functionality and associated datasets enhance and advance the possibilities of ecosystem research. The data help researchers to describe the climate at sites, explore climate events in the past and future, provide a basis for regional climate change assessments, and serve as input data for impact studies.

The official EURO-CORDEX data repository, as part of the ESGF federated storage infrastructure, changes over time mainly because more ensemble members are added by the international CORDEX consortium (for example more RCMs, version changes of RCMs, dataset updates, dynamical downscaling of previously unconsidered GCM, GCM experiment or climate scenarios) but there might also be withdrawal of RCM datasets. A local check-summed replica of the subset of the ESGF repository relevant for eLTER is maintained and updated long-term at the JSC (JSC, 2020) as a basis for the cyclic updates of the eLTER regional climate scenario data. The complete dataset will be updated and made available through B2SHARE and DEIMS-SDR when necessary, ensuring that the eLTER community continues to be provided with up-to-date regional climate scenarios. Previous versions will remain available via B2SHARE and DEIMS-SDR. It is likely therefore, that the number of ensemble members and time series will grow over time. The data product described in this paper is the first version, dated 1st August 2019.

Future versions of this data product will feature multiple time series of the same variable for individual sites after a bias adjustment will have been applied utilising quality-checked and error-corrected meteorological observations from those ecological research sites that make these data available through the DEIMS-SDR. Furthermore the eLTER information system might be connected directly with climate services such as the Copernicus Climate Data Store through dedicated APIs, making data retrievals for the ecological research sites even more efficient and flexible.

## Author contributions

SR led the writing of this manuscript and coordinated the project. KG was responsible for retrieving, handling, processing and extracting, visualising, documenting and staging the EURO-CORDEX RCM model data and helped with the writing and revisions of the manuscript. CW is responsible for DEIMS-SDR development, implemented the integration of climate data into the DEIMS-SDR and helped with repository handling. SA helped in setting up the B2SHARE data store and provided software tools and guidance to work with the data repository. The co-authors are part of the eLTER climate change projection or information management project teams and they all contributed to the writing, discussion and review of this manuscript.

**Competing interests**

The authors declare that they have no conflict of interest.

**Acknowledgements**

The eLTER EURO-CORDEX regional climate projections and DEIMS-SDR are products of LTER-Europe. eLTER was funded through the European Union's Horizon 2020 research and innovation programme (grant agreement no 654359). We acknowledge the World Climate Research Programme's Working Group on Regional Climate, and the Working Group on Coupled Modelling, the former coordinating body of CORDEX and responsible panel for CMIP5. We also thank the climate modelling groups for producing and making available their model output. We also acknowledge the Earth System Grid 355 Federation infrastructure, an international effort led by the U.S. Department of Energy's Program for Climate Model Diagnosis and Intercomparison, the European Network for Earth System Modelling and other partners in the Global Organisation for Earth System Science Portals (GO-ESSP). We also thank U. Schulzweida for the freely available CDO tool.

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

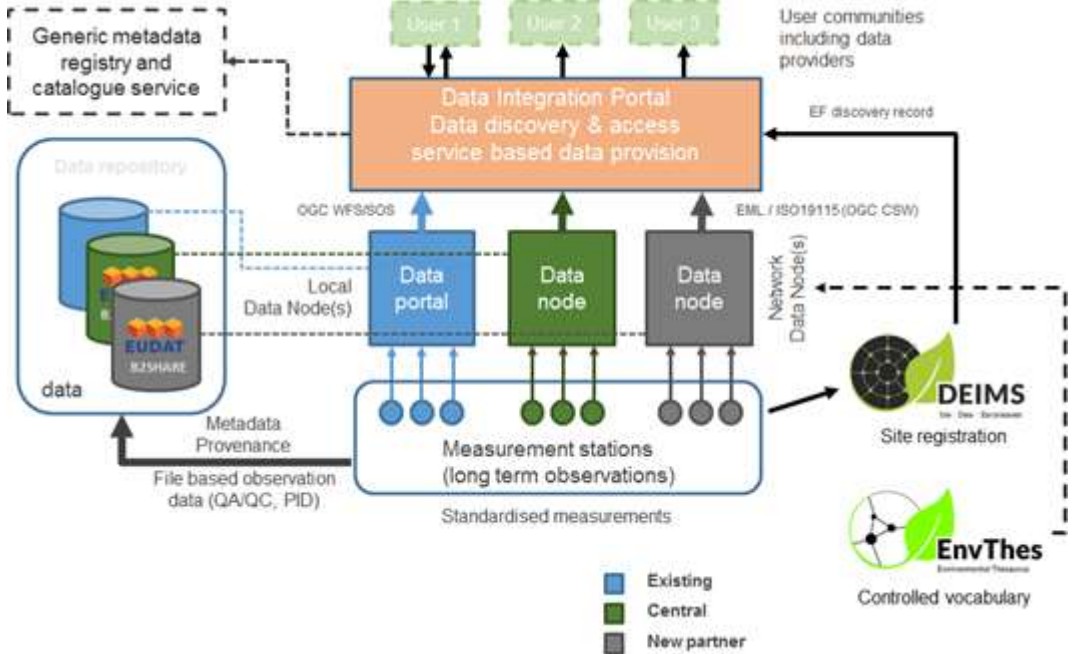


**Figure 1 Schematic Overview of the eLTER Information System components (source: eLTER Information System, 2020)**

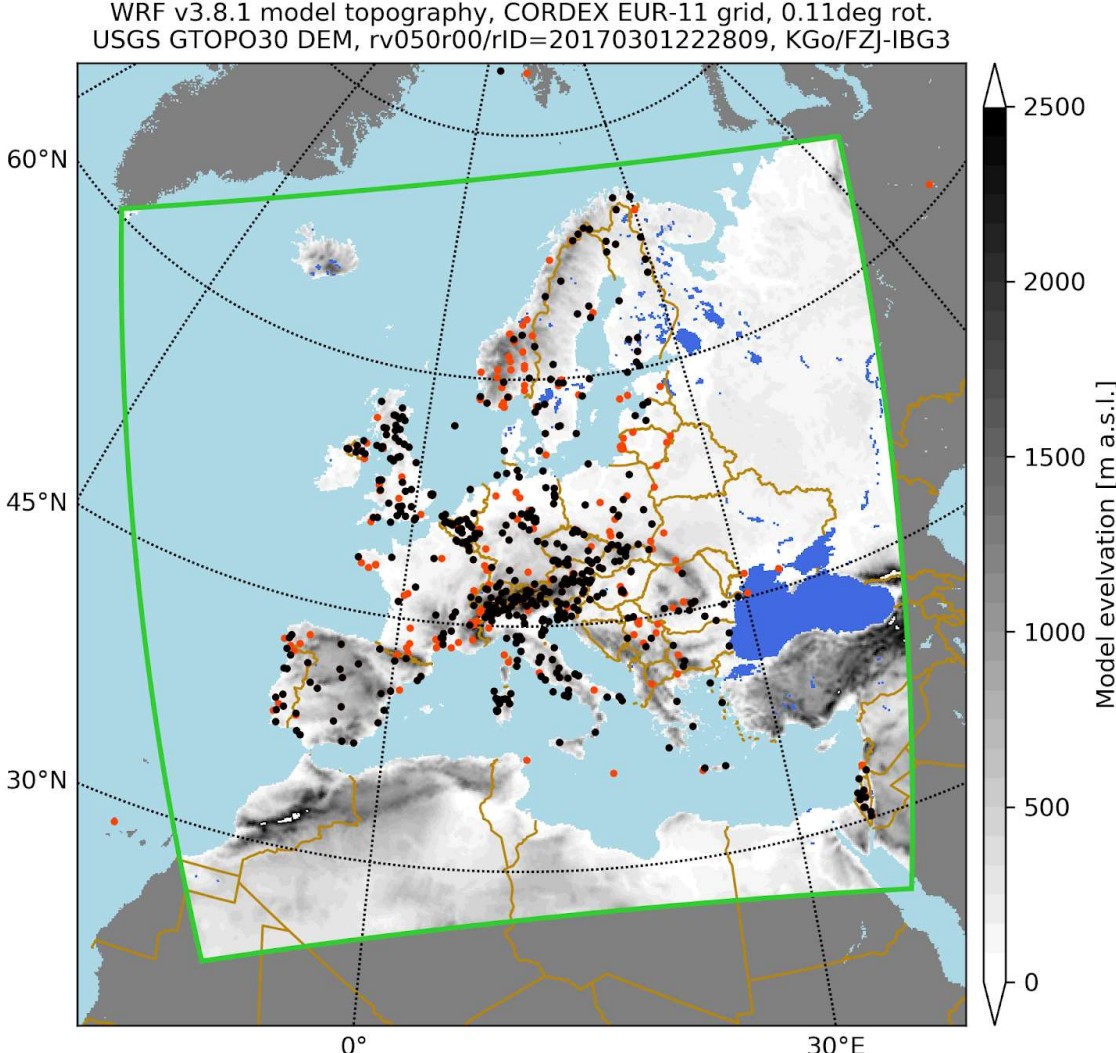

WRF v3.8.1 model topography, CORDEX EUR-11 grid, 0.11deg rot.
USGS GTOPO30 DEM, rv050r00/rID=20170301222809, KGo/FZJ-IBG3

**Figure 2 Overview of European ecological research site locations for which data are provided based on the EURO-CORDEX RCM ensemble. Dots: black: eLTER sites; red: other DEIMS-registered European LTER sites. Green line: Delimiter of the RCM focus domain as defined in the EURO-CORDEX project; EUR-11 grid: 424x412 grid elements. The topography in grey is taken from a WRF RCM (Knist et al., 2018) and based on the USGS GTOPO30 (USGS, 2020) global elevation dataset**


**a)**

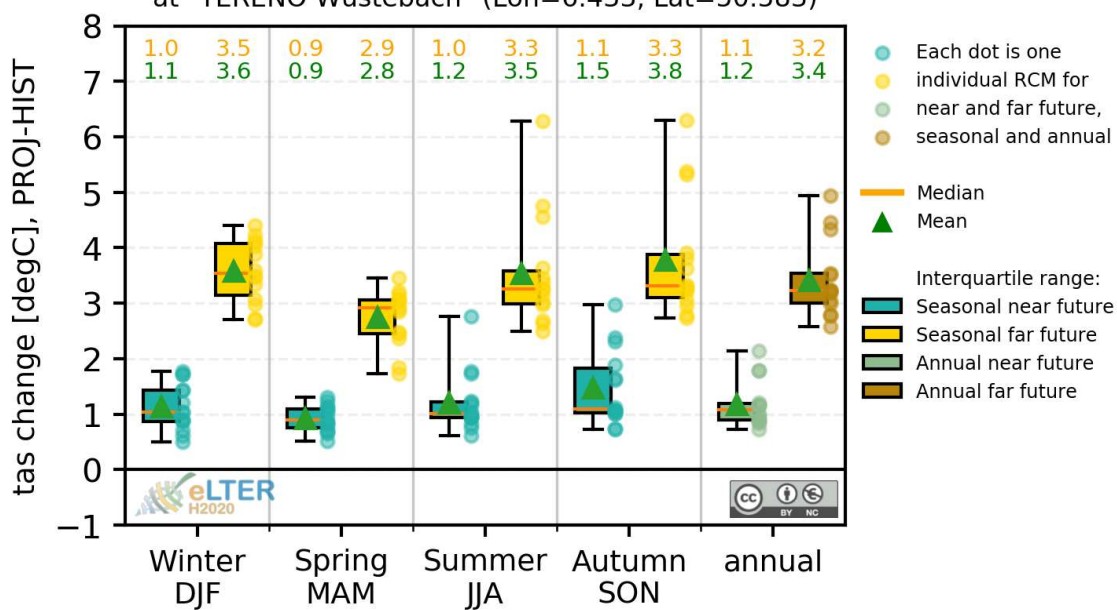

Projected Near-Surface Air Temperature (tas) Change
from EURO-CORDEX RCP85 EUR-11 RCM Ensemble (n=14)
at "TERENO Wüstebach" (Lon=6.433, Lat=50.583)

Differences of 30-year seasonal and annual means
Near future: 2021-2050 (PROJ1) minus 1971-2000 (HIST),
far future: 2069-2098 (PROJ2) minus 1971-2000 (HIST)

https://deims.org/9fe5a5d1-ccc0-41ab-b555-5ca44da24cd8
Versions: data=2018-02-07, processing=2019-05-21, plot=2019-07-29

b)

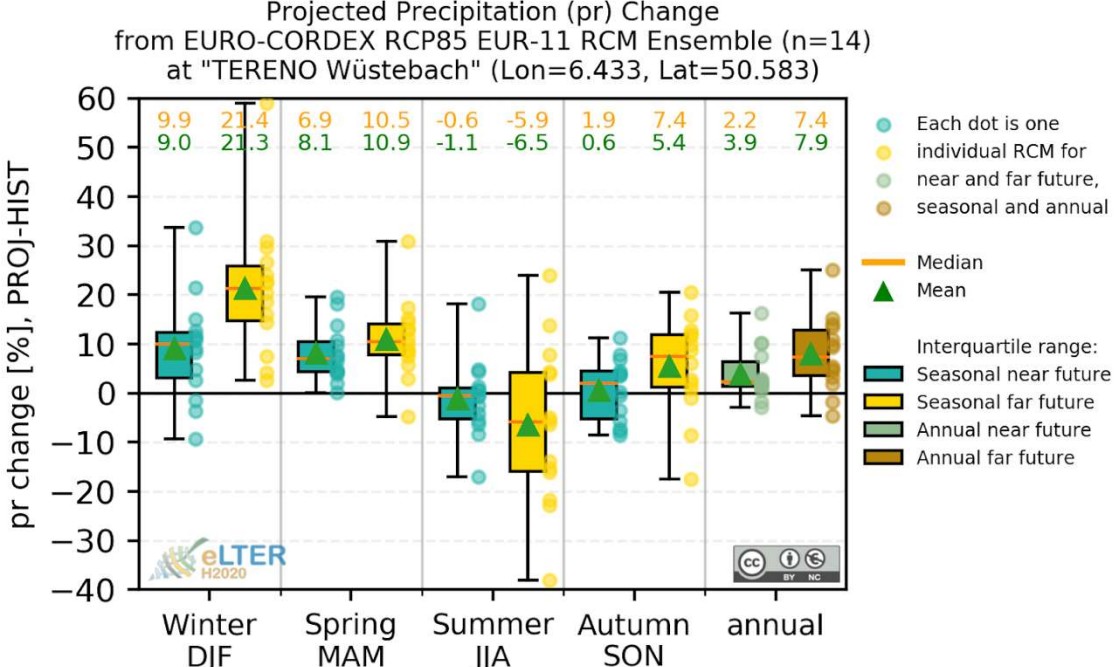

**Figure 3 Example eLTER EURO-CORDEX climate scenario data visualisation for one of the TERENO sites in Western Germany: (a) near surface air temperature (tas) and (b) total precipitation (pr)**

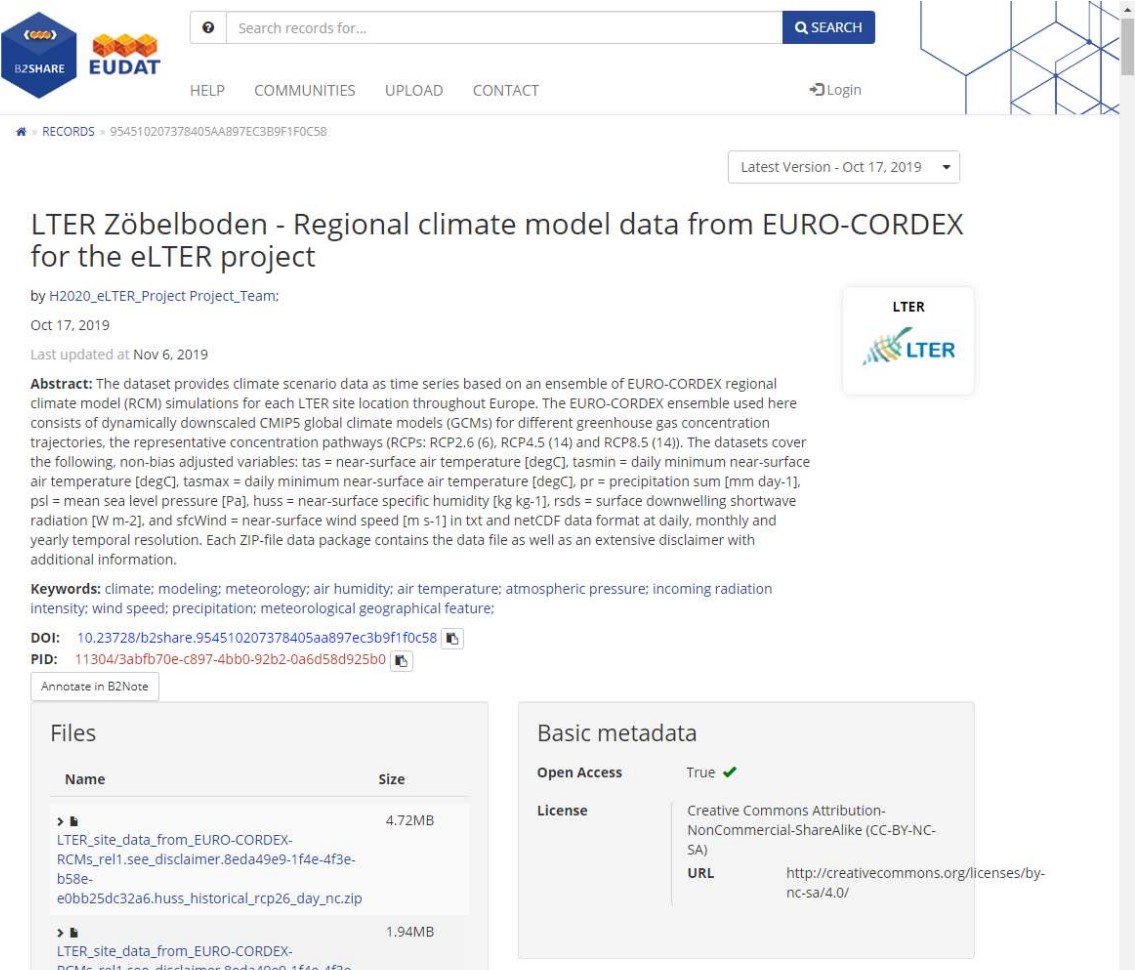

**Figure 4 B2SHARE record**

# Cairngorms (ECN site) - United Kingdom

DEIMS.iD: https://deims.org/5a04fee1-42aa-47e9-abfc-043a3eda12ac

## Basic Information

**Site Name:** Cairngorms (ECN site)
**Short name:** T12
**Country:** United Kingdom
**Web Address:** ECN Data Centre: Cairngorms
ECN Data Centre: Home
**Parent Site Name:** Cairngorms National Park LTSER - United Kingdom
**Site Manager:** Chris Andrews
**Operating Organisation:** UK Centre for Ecology & Hydrology
**Keywords:** eLTER catalogue
GLORIA
LTER Site
LTSER platform
Socio-ecological
Tea bag
TeaComposition
**Site Description:** UK Environmental Change Network (ECN) site. The Cairngorms site is located high in the Cairngorms, near Aviemore in Speyside, Scotland. The site lies on the North-Western flank of the Cairngorms encompassing the catchment of the Allt a' Mharcaidh (a site in the ECN freshwater network). It is part of the Invereshie and Inshriach National Nature Reserve, within the Cairngorms National Park, and covers some 10 km2. This is the first ECN site in the UK's sub-arctic zone and is an important link not only to other upland ECN sites but to alpine site in Europe and globally through the GLORIA network, and also to networks across the Arctic (SCANNET and INTERACT). The Cairngorms site has been used intensively for research since the ...
Show more ...
**Last modified:** 2020-03-30 03:03:54

## Photos

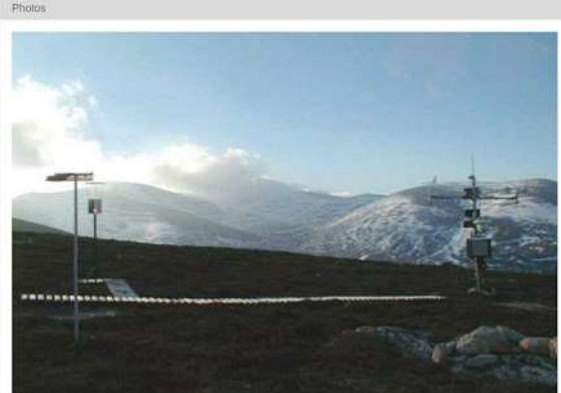

## General Characteristics, Purpose, History

**Site Status:** existing
**Year Established:** 1999
**Purpose:** Nature conservation, outdoor recreation and research. The site is ideally placed to monitor changes in: * Tree colonisation - Mature trees (principally Scots pine, Pinus sylvestris) at the site are confined to a relatively small part of the lower ground and there had been no significant regeneration over the last two centuries due to heavy grazing by deer, and burning. However, a reduction in deer grazing began around 20 years ago and colonisation by saplings is now widespread; *Climate change - The site straddles the zones of increasing winter precipitation and decreasing summer precipitation whilst there is also evidence of increasing windiness. The specialised arctic-alpine plant communities found at the site can be used to ...
Show more ...
**History:** The Cairngorms site has been used intensively for research since the 1970s. An Automatic Weather Station (AWS) has been operating at the site since 1984 and was used in the Surface Water Acidification Programme from 1984 to 1994. CEH and MI have used the site for long-term hydrological and snow studies for about 15 years. From 1997-1999 it was one of the ECOMONT (land use change in mountain areas of Europe) sites. The site joined the UK Environmental Change Network (UK LTER) in 1999.
**Research Topics:**
biology, ecology, biodiversity, species diversity, animal ecology, aquatic ecology, stream ecology, ecosystem ecology, plant ecology, vegetation dynamics, terrestrial ecology, forest ecology, ecosystem service, ecosystem function, chemistry, biogeochemistry, depositions chemistry, soil chemistry, water chemistry, atmospheric chemistry, geology, hydrology, meteorology, climatology, climate change, climate monitoring, environmental science, geography
**Parameters:**
dissolved organic carbon in soil, atmospheric parameter, air humidity, air temperature, ozone concentration, precipitation intensity, snow depth, incoming radiation intensity, net radiation irradiance, reflected radiation intensity, solar radiation, net solar radiation irradiance, total radiation irradiance, wind direction, wind speed, biological parameter, leaf area, flowering date, emergence date, plant development stage, plant cover, percent organic carbon, total carbon, total organic carbon, available phosphorus, inorganic carbon content, percent carbon, soil pH, dissolved organic phosphorus, ecosystem parameter, floristic diversity, species composition, plant species composition, species abundance, species presence, birds presence, bryophytes presence, soil solute amount, species turnover, diversity index, landscape parameter, land cover, land use, soil parameter, soil acidity, soil bulk density, carbon-to-nitrogen ratio, cation exchange capacity, soil moisture field capacity, soil temperature, soil solute amount, soil solution concentration, dissolved organic nitrogen in soil, thickness of soil horizon, total organic carbon, water parameter, water acidity, conductivity, runoff amount, water level, stage height, water quality, water table, water volume

## Geographic

Size: 1000.00ha
**Elevation (average):** 766.00msl
**Elevation (min):** 320.00msl
**Elevation (max):** 1110.00msl

## Affiliation and Network Specific Information

**Affiliation:** ILTER ✓
LTER Europe ✓
UK ECN ✓ (LTER_EU_UK_055)
INTERACT ✗ (https://eu-interact.org/field-sites/ecn-cairngorms/)
**Projects:**
ALTER-Net, Critical Zones, SCANNET, UK Environmental Change Network, UK Eutrophying and Acidifying atmospheric Pollutants (UKEAP) network, eLTER catalogue, eLTER (H2020), Teabag

## Download

Site information [.json]
Centroid coordinates [.shp] [.kml]
Bounding Box [.shp] [.kml]
Boundaries [.shp] [.kml]

▸ Site Details

## Related Resources

18 dataset(s) in total:
Acoustic bat data: 2014
Cairngorms: Ecosystem services variables from the UK Environmental Change Network (ECN)
Cairngorms: UK Environmental Change Network (ECN) bat data: 1993-2015
Cairngorms: UK Environmental Change Network (ECN) bird data: 1995-2016
Cairngorms: UK Environmental Change Network (ECN) butterfly data: 1993-2015
Cairngorms: UK Environmental Change Network (ECN) frog data: 1994-2015
Cairngorms: UK Environmental Change Network (ECN) Precipitation Chemistry data: 1992-2015

1  2  3  Next ›  Last »

▸ Additional data

There is **EURO-CORDEX** climate scenario data available for this site:

| | | | |
|---|---|---|---|
| Projected Near Surface Specific Humidity Change | RCP26 | RCP45 | RCP85 |
| Projected Precipitation Change | RCP26 | RCP45 | RCP85 |
| Projected Sea Level Pressure Change | RCP26 | RCP45 | RCP85 |
| Projected Surface Downwelling Shortwave Radiation | RCP26 | RCP45 | RCP85 |
| Projected Near-Surface Wind Speed | RCP26 | RCP45 | RCP85 |
| Projected Near-Surface Air Temperature | RCP26 | RCP45 | RCP85 |
| Projected Daily Maximum Near-Surface Air Temperature | RCP26 | RCP45 | RCP85 |
| Projected Daily Minimum Near-Surface Air Temperature | RCP26 | RCP45 | RCP85 |

You can also download the entire climate scenario dataset for this site from the EUDAT B2SHARE data store.

**Figure 5 DEIMS-SDR metadata viewer with the access to the climate information at the bottom**

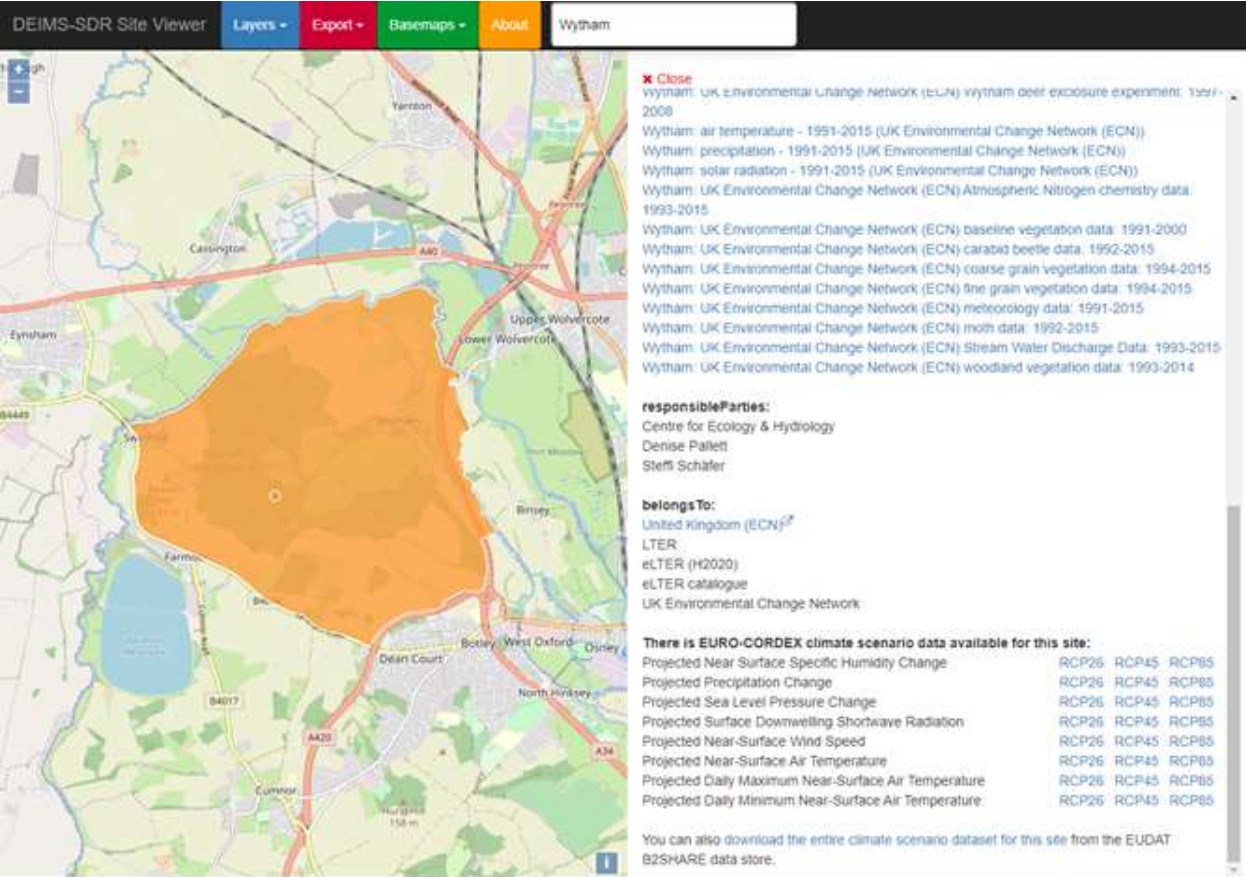

**Figure 6 Data Discovery on the DEIMS-SDR site map**

595

| Variable name | Description | Temporal aggregation | Unit |
|---|---|---|---|
| tas | Near-surface air temperature | Daily mean | °C |
| tasmin | Near-surface air temperature | Daily minimum | °C |
| tasmax | Near-surface air temperature | Daily maximum | °C |
| pr | Precipitation | Daily total | mm |
| psl | Sea level pressure | Daily mean | hPa |
| huss | Near-surface specific humidity | Daily mean | kg kg-1 |
| rsds | Surface downwelling shortwave radiation | Daily mean | W m-2 |
| sfcWind | Near-surface wind speed | Daily mean | m s-1 |

**Table 1** Meteorological variables provided in dataset release 1 (depending on user demand, this list can be expanded).

600