# Peer review of "A Climate Service for Ecologists: Sharing pre-processed EURO-CORDEX Regional Climate Scenario Data using the eLTER Information System"

_Earth System Science Data, 2020_

## Referee Comment (RC1) · Bruno Petriccione (Referee) · 15 Nov 2020

Quality of the paper is very high, with an elevated level of accuracy and completeness. Data availability, as well explained, will save a lot of time and will increase quality of research for many ecologists. Thanks to this paper, general scientific community will know as long term ecological research is valuable and well organised also at European level.

Non specific comments.

Only one technical coorection: delete "(" at row no, 139.

Good luck, Bruno Petriccione

---

## Referee Comment (RC2) · Annalisa Minelli (Referee) · 2 Jan 2021

General Comments:

The overall quality of the work is high and in my opinion it is highly relevant to the ecological scientific community since:

- it is meant for ecologists working with LTER data, and since the LTER community has a reference "place" for data (DEIMS-SDR), it simplifies a lot the researchers' life to have a unique hub where to retrieve all information available on LTER sites;

- even if elaborations have been performed only for some variables, these variables are the most common used in ecological studies. Other variables are however accessible and well referenced;

- even if "raw" data is already published, this publication is a re-work of existing data with an original and well documented processing.

Specific Comments:

The choice of B2SHARE platform for data sharing is correct and perfectly compliant with FAIR data management principles, especially in terms of accessibility.

Data temporal granularity is appropriate for the research field data are supposed to serve.

Data uncertainty is meticulously taken into account and explained.

The file naming is systematic and well documented, I only noticed that in the ".zip" files downloadable from B2SHARE there is a "see_disclaimer" between type of data (e.g. "RCM") and DEIMS UUID. How including this statement in the zipped file name is relevant?

A curiosity: it is already planned, as far as you know, a similar service for LTER sites outside Europe? The project aims are relevant and its wider implementation would be an added value to the research of many ecologists around the world.

Technical Corrections:
There's a "(" at row 129 that must be deleted.

---

## Author Response (AR2)

**Response to Referee Comments**

We thank Bruno Petriccione and Annalisa Minelli for their thoughtful and helpful reviews.

5   **Referee 1 – Bruno Petriccione**

We thank Bruno Petriccione for his very positive review.  We are pleased that he feels that this work will be valuable for ecologists and for his positive comments about the data description.

**Referee 1- Comment 1**:

10   *"Only one technical coorection: delete "(" at row no, 139."*

We don't see a "(" at row 139.  We think that this may be the "(" indicated by reviewer 2 at row 129.  This has been corrected, many thanks for highlighting it.

15   **Referee 2 – Annalisa Minelli**

We thank Annalisa Minelli for her thoughtful review.  We are pleased that she recognises the value of this work to LTER researchers.  We agree that integrating these data within DEIMS-SDR will simplify things for LTER researchers.  As the referee has highlighted we chose to focus on eight variables.  These were chosen as they are the most commonly used variables to assess regional climate change, extremes and as input data to impact assessment and modelling.  We recognise that other

20   variables are available and that CORDEX data are available elsewhere, as referenced in the manuscript.  We are pleased that Annalisa Minelli recognises the value of providing this as a bespoke data resource for LTER researchers who will not previously have had access to these data and hope that this will increase their capabilities for long term ecological research.

We are gratified that the data description, our chosen temporal granularity and our comments about data uncertainty, as well as the choice of B2Share for data sharing, are clear and acceptable.

**Referee 2- Comment 1**:

*"The file naming is systematic and well documented, I only noticed that in the ".zip" files downloadable from B2SHARE there is a "see_disclaimer" between type of data (e.g. "RCM") and DEIMS UUID. How including this statement in the zipped file name is relevant?"*

30   As noted by the referee, we have implemented a strict naming convention for the files and included as much information within the file name itself (described in section 4 of the manuscript).  We felt this was essential given the large number of files that are involved.  We also wanted to be sure that the user would have access to technical information about each dataset, including limitations, uncertainties and constraints to using the data.  This information is provided as a 'disclaimer' within each data

granule and is provided in section 7 of this manuscript. We have highlighted this in the zip file to ensure that users can easily
35 find this information and take it into account.

**Referee 2- Comment 2**:

*"A curiosity: it is already planned, as far as you know, a similar service for LTER sites outside Europe? The project aims are relevant and its wider implementation would be an added value to the research of many ecologists around the world."*

40 This work was funded as part of the eLTER project - through the European Union's Horizon 2020 research and innovation programme. We have therefore concentrated on LTER sites in Europe. We agree that it would be useful to extend this to other LTER sites outside of Europe that are documented in DEIMS-SDR. We will also look for further funding opportunities to expand this data resource in content (more ensemble members and experiments) and functionality (bias adjustments). The main limitation for expanding the dataset beyond Europe lies in the availability of regional climate model data. Despite the
45 fact that CORDEX is a global World Climate Research Programme effort with global coverage by different regional climate model domains, the availability of ensemble data varies and is especially high for the European model domain as used in the current dataset. Technically however, this would be feasible.

**Referee 2- Comment 2**:

50 *"There's a "(" at row 129 that must be deleted."*

We thank the referee for highlighting this and have corrected it.

**A Climate Service for Ecologists: Sharing pre-processed EURO-CORDEX Regional Climate Scenario Data using the eLTER Information System**

Susannah Rennie[1], Klaus Goergen[2,3], Christoph Wohner[4,5], Sander Apweiler[6], Johannes Peterseil[4], John Watkins[1]

[1]UK Centre for Ecology & Hydrology, Lancaster Environment Centre, Library Avenue, Bailrigg, Lancaster, LA1 4AP, UK

60 [2]Institute of Bio- and Geosciences (Agrosphere, IBG-3), Research Centre Jülich, Wilhelm-Johnen-Straße, 52425 Jülich, Germany

[3]Centre for High-Performance Scientific Computing in Terrestrial Systems, Geoverbund ABC/J, 52425 Jülich, Germany

[4]Environment Agency Austria, Spittelauer Lände 5, 1090 Vienna, Austria

[5] Paris-Lodron University of Salzburg, Schillerstraße 30, 5020 Salzburg, Austria

Commented [RSC1]: This is not in response to a referee comment but we wanted to highlight that we have added additional affiliation information for author Christoph Wohner.

[revised manuscript text omitted]